# Common data elements for predictors of pediatric sepsis: A framework to standardize data collection

**Alishah Mawji**[1,2]*, **Edmond Li**[3], **Arjun Chandna**[4,5], **Teresa Kortz**[6], **Samuel Akech**[7], **Matthew O. Wiens**[2], **Niranjan Kissoon**[8], **Mark Ansermino**[1,2]

1 Department of Anesthesiology, Pharmacology & Therapeutics, University of British Columbia, Vancouver, British Columbia, Canada, 2 Centre for International Child Health, BC Children's Hospital Research Institute, Vancouver, British Columbia, Canada, 3 School of Population and Public Health, University of British Columbia, Vancouver, British Columbia, Canada, 4 Cambodia Oxford Medical Research Unit, Angkor Hospital for Children, Siem Reap, Cambodia, 5 Centre for Tropical Medicine and Global Health, University of Oxford, Oxford, United Kingdom, 6 Department of Pediatrics, Institute of Global Health Sciences, University of California, San Francisco, California, United States of America, 7 Kenya Medical Research Institute/ Wellcome Trust Research Programme, Nairobi, Kenya, 8 Department of Pediatrics, University of British Columbia, Vancouver, British Columbia, Canada

* alishah.mawji@bcchr.ca

**Data Availability Statement:** The common data element set, underlying data files, and all supplementary materials are available from the

## Abstract

### Background

Standardized collection of predictors of pediatric sepsis has enormous potential to increase data compatibility across research studies. The Pediatric Sepsis Predictor Standardization Working Group collaborated to define common data elements for pediatric sepsis predictors at the point of triage to serve as a standardized framework for data collection in resource-limited settings.

### Methods

A preliminary list of pediatric sepsis predictor variables was compiled through a systematic literature review and examination of global guideline documents. A 5-round modified Delphi that involved independent voting and active group discussions was conducted to select, standardize, and prioritize predictors. Considerations included the perceived predictive value of the candidate predictor at the point of triage, intra- and inter-rater measurement reliability, and the amount of time and material resources required to reliably collect the predictor in resource-limited settings.

### Results

We generated 116 common data elements for implementation in future studies. Each common data element includes a standardized prompt, suggested response values, and prioritization as tier 1 (essential), tier 2 (important), or tier 3 (exploratory). Branching logic was added to the predictors list to facilitate the design of efficient data collection methods, such as low-cost electronic case report forms on a mobile application. The set of common data

Pediatric Sepsis CoLab Dataverse (https://doi.org/
10.5683/SP2/02LVVT).

**Funding:** AC is supported by a Wellcome Trust
Doctoral Training Fellowship. SA was supported by
the Initiative to Develop African Research Leaders
(IDeAL) Wellcome Trust award (# 107769). TK
receives salary support from the National Institute
of Allergy and Infectious Diseases, USA
(K23AI1440.29).

**Competing interests:** The authors have declared
that no competing interests exist.

elements are freely available on the Pediatric Sepsis CoLab Dataverse and a web-based
feedback survey is available through the Pediatric Sepsis CoLab. Updated iterations will
continuously be released based on feedback from the pediatric sepsis research community
and emergence of new information.

## Conclusion

Routine use of the common data elements in future studies can allow data sharing between
studies and contribute to development of powerful risk prediction algorithms. These algo-
rithms may then be used to support clinical decision making at triage in resource-limited set-
tings. Continued collaboration, engagement, and feedback from the pediatric sepsis
research community will be important to ensure the common data elements remain applica-
ble across a broad range of geographical and sociocultural settings.

## Introduction

Sepsis is life-threatening organ dysfunction due to a dysregulated host response to infection
[1]. Despite a global trend of decreasing incidence and mortality, sepsis remains a major cause
of health loss worldwide and has an especially high burden in low-and-middle-income coun-
tries [2]. In recognizing the enormity of the global burden of sepsis in terms of mortality, mor-
bidity, and social and economic consequences, the World Health Assembly, the decision-
making body for the World Health Organization highlighted the urgent need for strengthened
efforts to identify, prevent, and treat sepsis [3].

Improved prioritization, coordination, and timely identification of critically ill (or at risk of
becoming critically ill) children has been recognized as cornerstone in the efforts to improve
sepsis outcomes [4,5]. Clinical decision making in these areas can be supported and enhanced
with a data driven precision health approach and patient-centred, personalised risk prediction
recommendations. The importance of powerful data driven prediction algorithms is under-
lined by the fact that they can be used by healthcare workers with less experience and training,
a common scenario in resource-limited settings where the burden of sepsis is highest. Early
identification of the at-risk patient can improve healthcare delivery by facilitating timely
administration of treatment before significant deterioration occurs, and early referral when a
higher level of care is required.

Progress in the adoption of this data driven approach to optimize health and healthcare has
been limited by the lack of robust, high quality sources of data and validated clinical outcomes
[6]. It is becoming increasingly clear that multi-center collaborations and combining data
from multiple settings are necessary to generate the highest level of evidence to guide clinical
practice [6]. However, the linking and aggregation of collected data are often limited by the
varying degrees of data compatibility across studies. One way to ensure meaningful data
exchange and compatibility into the future is to support harmonized use of a recommended
set of common data elements. A common data element is a combination of a precisely defined
question (predictor variable) paired with a specified set of clearly defined responses to the
question [7]. Common data elements can also be classified into tiers of importance to docu-
ment the relevance of each data element for future studies and to advise a minimum set of rou-
tinely collected data across studies [8]. By increasing consistency of data collected across
studies common data elements can facilitate data aggregation, meta-analyses, cross-study

comparisons, and simplify training operations to increase overall efficiency and quality of data collection [7].

The purpose of this study was to create a universally accepted living framework to increase standardization of data collection in future studies of pediatric sepsis prediction. The Pediatric Sepsis Predictor Standardization (PS2) working group, a subgroup within the Pediatric Sepsis CoLab, has collaborated to develop a set of common data elements for predictors of pediatric sepsis at the point of triage in resource-limited settings. Standardized outcomes are being addressed by a related working group [9,10]. Routine use of the framework within the pediatric sepsis research community would promote an environment of collaboration, improve compatibility of datasets across studies, and yield robust, high quality sources of data that can be used to develop powerful risk prediction algorithms and support clinical decision making.

## Methods

Ethics approval was obtained from the Child and Women's Research Ethics Board at the University of British Columbia (H17-01893). All expert contributors are members of the Pediatric Sepsis CoLab and voluntary involvement in the Delphi process implied consent.

### Pediatric sepsis predictor standardization (PS2) working group

The PS2 working group is a subgroup within the Pediatric Sepsis CoLab, a global network for collaboration and data sharing to address pediatric sepsis morbidity and mortality [11,12]. The PS2 working group is currently comprised of six experts in pediatric sepsis with representation from Canada, the United States of America, East Africa, and Cambodia; and clinical expertise including pediatric critical care, infectious disease management, and anesthesia.

### Study design

A preliminary list of potential predictor variables was compiled based on a systematic review [13] followed by a 5-step modified Delphi approach [14] to develop a set of common data elements relevant to prediction of pediatric sepsis at resource-limited health facilities. Standardized predictors, definitions, prompts, accepted values, and importance tiers were developed using four 60-minute conferences, separated by rigorous revisions based on literature review and expert opinion. An 80% threshold was used to define consensus. The final set of common data elements and guidelines for their use have been published to the Pediatric Sepsis CoLab's Dataverse [15], a data-sharing platform used by the CoLab to share resources and study data internationally among members.

**Systematic review.** A prior literature review conducted by our group [13] was used to assemble a preliminary list of candidate predictor variables. The systematic search was conducted using the Ovid MEDLINE database, and the search strategy included terms such as "child", "sepsis", "triage", "prediction", and "resource-limited settings". The complete search strategy is available in the S1 Appendix. The most up-to-date versions of published triage guidelines such as Integrated Management of Childhood Illness, Emergency Triage Assessment and Treatment, and the South African Triage Scale were also reviewed for candidate predictors. In addition, reference lists of included studies were screened for potential manuscripts which may have yielded additional candidate predictors. Lastly, case-report forms from large prediction studies in progress by collaborators and working group members were reviewed in order to capture currently used sepsis predictors [16–20]. The systematic review adhered to the Preferred Reporting Items for Systematic Reviews and Meta-Analyses (PRISMA) guidelines (S1 Checklist) [21].

A master list was created using Microsoft Excel to track each predictor variable and its frequency of use across studies. Variables were classified by field type (binary, continuous, categorical), and organized into seven core domains: patient characteristics/history, pregnancy/birth details, sociodemographic information, anthropometric data, vital signs, clinical signs and symptoms, and laboratory tests. The clinical signs and symptoms domain was further divided into seven sub-categories: respiratory, circulation/perfusion/dehydration, neurological, infection, gastrointestinal/genitourinary, malnutrition, and trauma. The working spreadsheet documenting the results of the systematic review and source studies for each predictor is available on the Pediatric Sepsis CoLab Dataverse [22].

**Delphi process.** A 5-step modified Delphi process was used to determine the final list of predictors, develop standardized definitions, response values and terminology for data collection; as well as to assign a tier of importance to each variable (Fig 1).

**Step 1:** Independent review of the preliminary list of potential sepsis predictors to comprise the core common data element set.

Members of the PS2 working group independently reviewed the master list of predictors and flagged predictors to exclude from the common data element set. The decision to exclude was based on considerations of the perceived predictive value of the candidate predictor at the point of triage, intra- and inter-rater measurement reliability, and amount of time and material resources required to reliably collect the predictor in resource-limited settings. Predictors with 100% consensus for exclusion were excluded. Predictors that did not reach this consensus agreement were discussed in Step 2.

**Step 2:** Consensus meeting to determine final list of predictors to comprise the common data element set.

Step 2 occurred as a 60-minute web conference during which predictors without unanimous exclusion consensus were openly discussed until consensus was reached. At that time, members were encouraged to suggest additional variables and reach consensus as a group on whether each of these variables should be included in the common data element set. By the end of the meeting, the list of included predictors for common data element set was agreed upon and finalized.

**Step 3:** Two web conferences to develop standardized definitions, prompts and responses for each variable.

For each predictor on the consensus list, standardized prompts and suggested response values were systematically constructed using a rigorous combination of literature sources and expert opinion. Operational definitions of predictors from the literature were consulted where available. Sources of operational definitions included study publications, case report forms from ongoing studies, and terminology standardization organizations such as SNOMED [23]. Where no operational definition was available from the literature, members of the working group constructed prompts that were easy to follow, objectively interpretable, and culturally compatible based on clinical and research experience. Predictors with relevance limited to a specific subset of participants (e.g. jaundice and bulging fontanelles in neonates) were also identified. Criteria for activating these predictors were selected based on accepted values in the literature and expert consensus.

The resultant list of predictors with standardized prompts and accepted responses was then circulated to the working group for individual voting. Prompts and responses flagged for review were discussed during a 60-minute web conference, after which the flagged items were further refined based on suggestions of the working group. A second round of voting and video conference was conducted to finalize the prompts and responses for each predictor.

**Step 4:** Independent voting on tier of importance for each predictor.

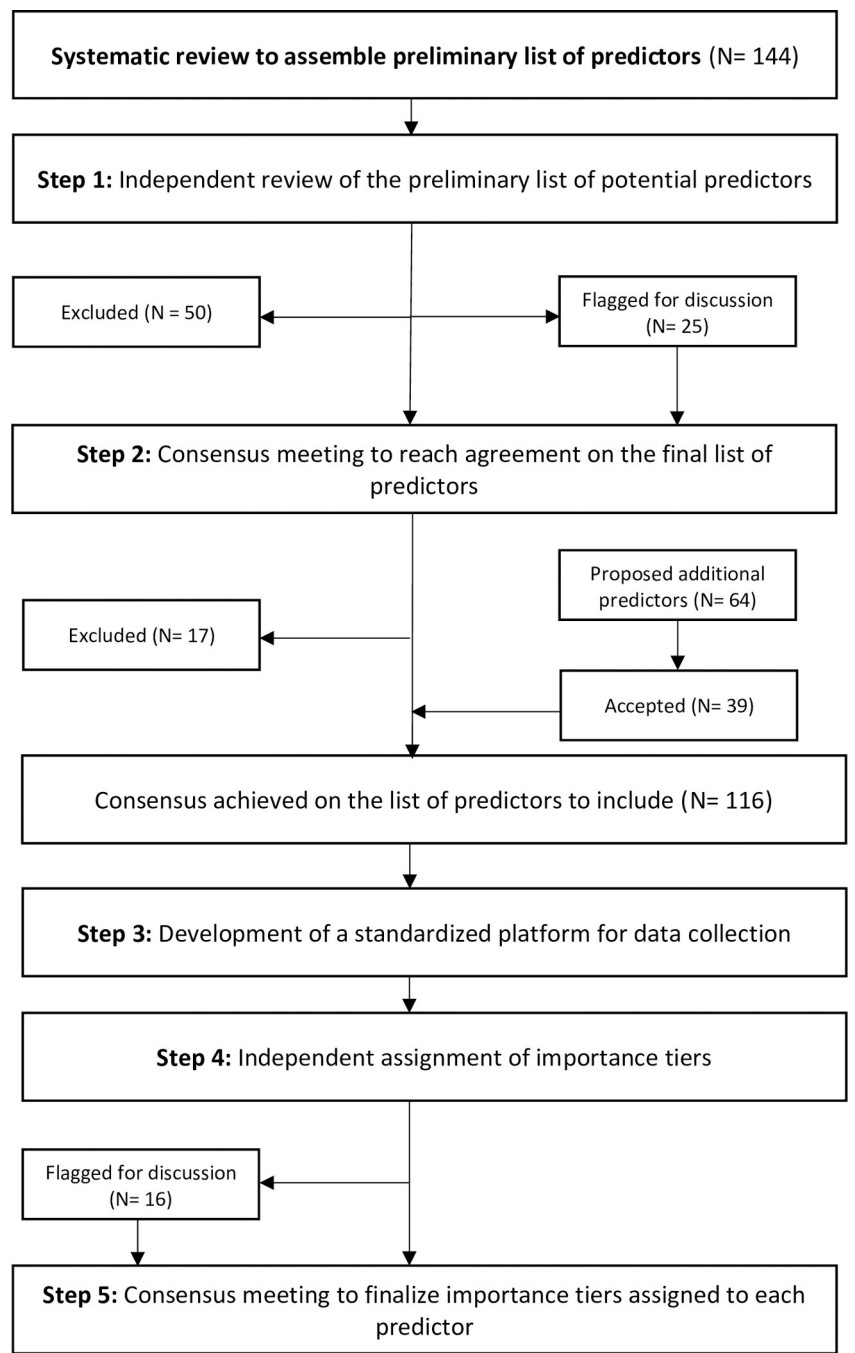

**Fig 1. Flowchart depicting the modified Delphi process used to develop the case report form for predictors of pediatric sepsis.**

To guide the inclusion of certain high-yield variables in future studies, each predictor was prioritized using a 3-point tiering scale based on importance from an epidemiological or data management point of view [24]. During Step 4, members of the PS2 working group independently assigned a tier of importance to each predictor. Predictors were ranked as Tier 1 when considered essential for all studies. These predictors comprise the consensus-based minimum data set to be collected across all future studies. Tier 2 predictors were defined as important

predictors that should be collected in relevant studies and when resources allow, and Tier 3 predictors were exploratory predictors or granular data necessary for a few specific studies within the field. Consensus agreement was defined as at least 80% among all members. Predictor variables that did not reach 80% agreement were discussed in Step 5.

**Step 5:** Consensus meeting to finalize importance tiers assigned to each predictor.

In Step 5, members engaged in a 60-minute web conference to openly discuss tiering of variables that did not reach 80% agreement until consensus was reached.

The results of the modified Delphi process were used to develop the common data element set that was then reviewed and approved by all members of the working group.

**Dataverse.** The set of common data elements and supplementary materials were published on the Pediatric Sepsis CoLab Dataverse, a secure platform that is internationally accessible but exhibits customizable permission control to protect sensitive data [15,22]. This Dataverse is the Pediatric Sepsis CoLab's main data-sharing platform and allows for data sharing among members with built in access and version control. To this end, patient-level datasets of predictors are also shared on Dataverse, alongside other general resources for CoLab members. Access to resources labelled as restricted files are available through a simple 5-minute online application [25]. Access to patient-level data may be obtained on a per-project basis by contacting the Pediatric Sepsis CoLab coordinator [15].

## Results

The final set of 116 common data elements and supplementary materials are published and freely available on the Pediatric Sepsis CoLab Dataverse [22].

### Final list of predictors

The literature review [13] yielded a preliminary list of 144 potential predictor variables (S1 Table) (Fig 1). Consensus for exclusion was attained for 50 of these variables during the independent review (S1 Table). The 25 variables that did not reach consensus for exclusion during the independent review were flagged for discussion in the consensus meeting (Step 2), and 17 of these variables were ultimately excluded (S1 Table). Most common reasons for exclusion were inadequate inter- or intra- rater measurement reliability (N = 31) due to subjectivity of the measures or the potential impact of recall bias on data integrity, and lack of perceived predictive value at the point of triage (N = 31) (S1 Table). Predictor variables were also excluded if information could be captured from a more detailed variable (N = 5). For example, the "born at home" predictor was captured as a response option in the more comprehensive "facility of birth" predictor. During the consensus meeting (Step 2), an additional 64 predictors were proposed for inclusion, 39 of which achieved consensus for inclusion (S2 Table). Consensus for inclusion in these guidelines was reached on a final list of 116 predictors (Fig 1). Predictors were grouped by categories and subcategories for ease of navigation of the guideline and to guide data grouping in future studies (Table 1).

### Features of the guidelines

**Importance tier.** Unanimous agreement was reached for all predictors ranked as Tier 1. Of the predictors ranked as Tier 2 and Tier 3, 16 required group discussion in order to reach consensus (S1 and S2 Tables). The common data element set consisted of 26 Tier 1 predictors, 66 Tier 2 predictors, and 24 Tier 3 predictors. The importance rank of certain variables depended on the geographical setting of the study and local infectious disease epidemiology. For example, a predictor exploring use of insecticide treated nets would rank as Tier 1 in

**Table 1. Schematic of the common data element guidelines with examples.**

| Recommended predictor | Tier | Standardized question or prompt on data collection platform/form | Possible values | Notes |
|---|---|---|---|---|
| | | *Clinical Signs and Symptoms*[a] | | |
| | | *Infection*[b] | | |
| | | *NEONATAL JAUNDICE branching logic*[c] | | |
| **Neonatal Jaundice (observed)**[d] | **1**[e] | **Does the child have yellowed skin?** | **Yes/no** | |
| IF: neonatal jaundice = yes[f] | | | | |
| Phototherapy needed? | 2 | Ask: "Did the child need light therapy for yellow skin?" | Yes/no/unknown | |
| Bilirubin measurement | 2 | | | See Laboratory section for details. |
| | | *RASH branching logic* | | |
| Rash (observed) | 2 | Does the child have a rash? | Yes/no | |
| IF: rash (observed) = yes | | | | |
| Localized rash (observed) | 2 | Is the rash localized? | Yes/no | |
| IF: localized rash (observed) = yes[g] | | | | |
| Rash body part (observed) | 3 | Record the body part where the rash is localized. | Scalp/face/neck/torso/left upper limb/right upper limb/left lower limb/right lower limb/perineal/other | |

[a]Predictor category.

[b]Predictor subcategory.

[c]Heading for branching logic tree.

[d]Tier 1 variables are bolded.

[e]Blue cells indicate variables applicable to young infants ($<$ 2 months).

[f]Red cells indicate first order branching logic conditions that trigger subsequent variables.

[g]Yellow cells indicate second order branching logic conditions that trigger subsequent variables.

malaria endemic regions and Tier 2 in most other settings (Table 2). Instructions explaining the variation in tier assignment of such predictors have been included in the guidelines.

**Standardized prompts.** A standardized prompt was constructed for each variable (Table 2). The prompts serve to standardize the syntax and operational definitions of predictors, and when applicable, the phraseology to be used by the research team during data collection. The prompts were carefully crafted with terminology comprehensible to respondents and language appropriate for the study setting. For variables captured by means of clinical

**Table 2. Sample predictors as presented in the common data element guidelines.**

| Recommended predictor | Tier | Standardized question or prompt on data collection platform/form | Possible values | Notes |
|---|---|---|---|---|
| | | *Sociodemographic data* | | |
| Bed-net use (reported) | 2 | Ask: "How often do you use insecticidal (bug resistant) bed nets for the child?" Read out options. | Never/rarely (<1 per week)/ sometimes (1–3 times/week)/often (4–6 times/week)/always/unknown | Consider tier 1 for malaria endemic regions, or if variable is otherwise of interest to the study. |
| | | *Clinical Signs* | | |
| | | *Circulation/perfusion/hydration* | | |
| **Capillary refill time $>$ 3 seconds (upper limb)** | **1** | **Apply pressure to a thumb or finger for 3 seconds to blanch it. Does it take more than 3 seconds to return to original pink color after you let go? (See SOP).** | **Yes/no** | **See SOP for details. 3 second cut-off shown to be applicable for all pediatric age groups (8).** |

examination, prompts also provided instructions to standardize measurements. Additional enhancements to the prompts may be required following field testing in various locations and following translation.

**Possible values.** A suggested set of response values are provided for each variable to facilitate compatibility between shared data sets (Table 2). The values can be used for data cleaning and to enhance opportunities for data linking.

**Notes.** An additional section is added for each variable that allows for communication of any special suggestions for using a predictor, rationale for how a predictor is presented, or other information that is likely to be of interest to users of the common data element set (Table 2).

**Branching logic.** Branching logic is used where applicable to avoid redundant questions, elucidate complex history concepts in a standardized manner, and expedite data collection (Table 3). Each branching logic tree is presented under a single heading for clarity (Table 1). The efficiency offered by branching logic can be maximally harnessed by using an electronic data collection interface rather than traditional paper methods [17].

**Standard operating protocols.** Standard operating protocols have been developed to provide additional standardization in variables with more involved physical exam procedures, such as anthropometric measurements and vital signs. These are available as supplementary material alongside the common data element guidelines [22].

**Feedback survey.** A web-based survey is available through the Pediatric Sepsis CoLab for researchers to propose new candidate predictors or suggest modifications to existing variables [26]. Feedback will be taken into account as input into future repetitions of the Delphi process to process to determine inclusion in updated iterations of the common data element set.

## Discussion

### Summary

Standardized data collection methods across pediatric sepsis prediction studies are lacking. As a step toward increasing consistency and compatibility for aggregation of future datasets, a multi-disciplinary team of experts in pediatric sepsis collaboratively generated a set of common data elements and guidelines for inclusion in case report forms or other data collection methods. Even when used within a single study, common data elements can provide consistency and efficiency in establishing data collection infrastructure and minimize variability in training and implementation [7]. Routine use of this standardized framework in future studies can increase

**Table 3. Sample branching logic to elucidate diarrhea timeline as per World Health Organization definitions.**

| Recommended predictor | Tier | Standardized question or prompt on data collection platform/form | Possible values | Notes |
|---|---|---|---|---|
| GI/GU | | | | |
| *DIARRHEA branching logic* | | | | |
| **Diarrhea (reported)** | **1** | **Ask: "Does the child currently have more than 3 loose stools a day?"** | **Yes/no/ unknown** | **Cut-offs of 3 loose stools are chosen based on WHO definition for diarrhea.** |
| IF: diarrhea (reported) = yes | | | | |
| **Persistent Diarrhea (reported)** | **1** | **Ask: "Has your child been passing loose stools for more than 2 weeks?"** | **Yes/no/ unknown** | **Cut-offs of 2 weeks based on WHO definition for persistent diarrhea.** |
| IF: persistent diarrhea (reported) = no | | | | |
| Dysentery (reported) | 2 | Ask: "Has the child had blood in their stools since getting loose stools?" | Yes/no/ unknown | Dysentery only asked for non-persistent diarrhea as per WHO definition. |

the quality and reproducibility of research and results in the field of pediatric sepsis, and contribute to development of well validated risk prediction algorithms to guide clinical care. Thus far, the framework has been implemented in the development of digital case report forms (mobile applications) for two independent studies of pediatric sepsis prediction [16–19].

## Adoption and use

The collective effort of a diverse group of expert clinician-researchers in the field of pediatric sepsis provided validation of the guidelines and ensured conformity with the current best standards in structured data curation. The broad representation of the expert panel will ensure applicability across a wide range of geographical settings and healthcare systems. The common data element set is meant to serve as the first iteration of a living guideline that will ultimately become the product of collaborative effort and continued engagement from the pediatric sepsis research community. In addition, the systematic review and Delphi process will be repeated every two years to capture new findings. Updated iterations of the common data element set will be published periodically based on contributions received through the feedback survey and emergence of new information.

## Limitations

Although the PS2 working group is comprised of expert clinical researchers diverse with respect to geographic location and healthcare setting, the group is not representative of all experts in pediatric sepsis across all geographic locations. It is possible that some variables excluded may in fact be important for prediction studies in select geographical or clinical settings. Further, the tier assignments have potential to be improved based on feedback from experts in geographic settings outside of those represented in the PS2 working group. Our decisions of inclusion and tier assignments were made in consideration of a delicate balance of achieving sufficient data quantity while avoiding diminished data quality due to study participant or staff fatigue. In future guideline iterations, these decisions will be improved upon based on feedback received from the pediatric sepsis community and new information in the literature.

As with the variability in predictor inclusion and importance, the applicability of the standardized prompts and responses may also vary among different geographical or healthcare settings. Additional response options may become available in specific locations and the availability or acceptability of measuring instruments can differ across facilities. In these cases, common data elements would have to be adapted to ensure relevance to the specific setting in which the data are being collected. We attempted to maximize applicability by utilizing culture-neutral phraseology for history questions and consulting international guidelines for standard operating procedures in clinical examination. Nonetheless, data sets from different settings will not be absolutely and completely consistent in actual practice. Despite this limitation, the use of common data elements can maximize compatibility between datasets and increase the quality of data collected within each dataset.

Lastly, despite the widespread use of the Delphi process as a best-practice method for consensus-based research, there are limitations inherent in its methods. The strength of Delphi methodology lies in its iterative process of independent opinion sharing and group feedback; it is commonly used in health care to gain expert consensus on a topic that may not be amenable to data driven methods of evidence generation [14]. The choice of repeatedly capturing independent opinions followed by active group discussions allowed for each voting round to be conducted anonymously and without potential influence from the opinion of others. The variables that did not reach unanimous agreement during anonymous voting were subsequently

discussed amongst the PS2 working group and were susceptible to potential bias introduced by group interpersonal dynamics. Nevertheless, active discussions promoted the use of collaborative thinking, generation of new ideas and solutions, and consideration of different perspectives to reach consensus on variables. To this end, the feedback survey is an extension of this collaborative modified Delphi process that involves the pediatric sepsis research community beyond the PS2 working group.

## Conclusion

Using a systematic review and modified Delphi approach, we developed a set of common data elements prioritized as essential (Tier 1), important (Tier 2), or exploratory (Tier 3) to serve as a standardized framework for future studies of pediatric sepsis prediction. This will enable collaboration and coordination in the pediatric sepsis research community and strengthen the quality of risk prediction models for pediatric sepsis to support clinical decision making at the point of triage.

## Supporting information

**S1 Checklist. Completed PRISMA checklist.**
(PDF)

**S1 Appendix. Search strategy for systematic review.**
(PDF)

**S1 Table. Preliminary list of predictor variables compiled through literature review (N = 144).**
(PDF)

**S2 Table. Additional predictor variables proposed in Step 3 (N = 64).**
(PDF)

## Acknowledgments

We would like to thank contributing members of the Pediatric Sepsis CoLab for their time, effort, and enthusiasm towards development of the common data element guidelines.

## Author Contributions

**Conceptualization:** Alishah Mawji, Arjun Chandna, Teresa Kortz, Samuel Akech, Matthew O. Wiens, Niranjan Kissoon, Mark Ansermino.

**Data curation:** Alishah Mawji, Arjun Chandna, Teresa Kortz, Samuel Akech, Matthew O. Wiens, Niranjan Kissoon, Mark Ansermino.

**Methodology:** Alishah Mawji, Edmond Li.

**Project administration:** Alishah Mawji.

**Supervision:** Edmond Li, Mark Ansermino.

**Writing – original draft:** Alishah Mawji.

**Writing – review & editing:** Edmond Li, Arjun Chandna, Teresa Kortz, Samuel Akech, Matthew O. Wiens, Niranjan Kissoon, Mark Ansermino.

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
