## [Decision Letter · Decision Letter 0]

1 Apr 2021

PONE-D-20-35664

Common data elements for predictors of pediatric sepsis: A framework to standardize data collection

PLOS ONE

Dear Dr. Mawji,

Thank you for submitting your manuscript to PLOS ONE. After careful consideration, we feel that it has merit but does not fully meet PLOS ONE’s publication criteria as it currently stands. Therefore, we invite you to submit a revised version of the manuscript that addresses the points raised during the review process.

We look forward to receiving your revised manuscript.

Kind regards,

Valérie Pittet, PhD

Academic Editor

PLOS ONE

Journal Requirements:

3. Please provide additional details regarding participant consent to collect personal data, including email addresses, names, or phone numbers. In the Methods section, please ensure that you have specified how consent was obtained and how the study met relevant personal data and privacy laws. If data were collected anonymously, please include this information.

[AC is supported by a Wellcome Trust Doctoral Training Fellowship. SA was supported by the Initiative to Develop African Research Leaders (IDeAL) Wellcome Trust award (# 107769).]

 [The authors received no specific funding for this work.]

5. We noted in your submission details that a portion of your manuscript may have been presented or published elsewhere. [Figure 1 (the systematic review process) was also published in our related manuscript. It is not dual publication because are just using the results of a literature review we conducted in the previous publication for a new purpose. This is further detailed in the cover letter. ] Please clarify whether this publication was peer-reviewed and formally published. If this work was previously peer-reviewed and published, in the cover letter please provide the reason that this work does not constitute dual publication and should be included in the current manuscript.

6. Please ensure that you refer to Figure 1 in your text as, if accepted, production will need this reference to link the reader to the figure.

7. Please include captions for ALL your Supporting Information files at the end of your manuscript, and update any in-text citations to match accordingly. Please see our Supporting Information guidelines for more information: http://journals.plos.org/plosone/s/supporting-information.

Reviewers' comments:

Reviewer's Responses to Questions

**Comments to the Author**

1. Is the manuscript technically sound, and do the data support the conclusions?

Reviewer #1: Yes

Reviewer #2: Yes

Reviewer #3: Yes

2. Has the statistical analysis been performed appropriately and rigorously? 

Reviewer #1: N/A

Reviewer #2: Yes

Reviewer #3: Yes

3. Have the authors made all data underlying the findings in their manuscript fully available?

Reviewer #1: Yes

Reviewer #2: Yes

Reviewer #3: Yes

4. Is the manuscript presented in an intelligible fashion and written in standard English?

Reviewer #1: Yes

Reviewer #2: Yes

Reviewer #3: Yes

5. Review Comments to the Author

Reviewer #1: The article presents a collection of predictors of pediatric sepsis sepsis and describes the process how this set was selected. The selection is based on expert’s opinions, not analysing data. The aim is to make the selected data elements as a standardised way to collect pediatric sepsis data. This would increase compatibility across different studies on the topic. The topic of the study is definitely important and results have a clear impact.

The research methodology is based on systematic literature review and domain experts’ knowledge through Delphi method. Finally, 116 data elements were accepted. The methodology is very well described and suitable for the study. However, the article does not describe whether any data collection has already done. Even a limited evaluation by using existing data would clearly improve the study.

Naturally, predictive power of the data items is difficult to measure since it would require a large amount of data and it also depends on the research questions at hand. If any data is already available, some data quality measures could have been computed. These measures could include basic descriptive statistics such as the number of observations per patient, missing values, correlations, and distributions. This statistical validation could have been an additional step in used methodology but it would also be an additional validation step for the defined set of data items.

The manuscript mentions that the aim is to facilitate the design of efficient data collections tools. Could you explain better what kind of tools these could be? In Conclusion, risk prediction tools are also mentioned. Are these two different sets of tools?

Other comments:

-Two tables S2 and S3 have the same title “Common data elements for predictors of pediatric sepsis: A framework to standardize data collection”

-On these tables a reason for exclusion can be “low perceived predictive values”. Is also based only on expert’s opinions or statistical analysis?

Reviewer #2: This is a very well written and important manuscript. The concept is well-defined and explained in the paper. There are some areas of redundancy that I think can be improved on (discussed more below).

Abstract:

- I really like how the amount of time and material resources to reliably collect the predictor in resource-limited settings was considered -- this is extremely important.

- Conclusion line 72: replace "remains" with "remain"

Introduction:

- Lines 102 and 103: Please standardize how you present "data-driven" vs "data driven", as you spell it with and without a hyphen.

- Line 105: Specify which settings you are referring to here. Low-resource settings?

- Line 106: Improving healthcare delivery seems vague here compared to the following two examples of timely treatment and early referral. Perhaps clarify a bit or omit.

- Line 113: Consider adding a source here.

- Lines 121-123: Use of commas is confusing here, and it ends up being a run-on sentence. I would use ";" to help organize this sentence.

- Line 125: For those unfamiliar with the PS2, I would suggest adding "a subgroup within the Pediatric Sepsis CoLab" after workgroup here. I know you go into more detail in the methods, but I would at least add that here in the Intro.

- Last paragraph: Clarify what the objective of this current paper is. Is it the first sentence? I would suggest making this more clear by stating "the objective of this study" or something like that.

Methods:

- Objective section: This seems a bit repetitive with the objective paragraph in the Intro. Perhaps include this there, instead, and cut where redundant?

- Line 173: Please keep past/present tense consistent throughout as past tense. Here please say "may have yielded".

- Line 174: Also, keep in either active or passive voice. This whole paragraph is in passive voice but the last sentence switches to active voice.

- Lines 188-190: Consider here the use of ";" rather than "," as otherwise this sentence is hard to follow.

- Lines 231-232: For those unfamiliar with the Delphi method, perhaps clarify what "80% threshold required for consensus". Does that mean that 80% of experts had to agree that it was important enough to be included?

- IRB approval was not discussed and should be included.

- If possible to add in a little bit about the experts, like what countries they represent, that would be very interesting.

Results:

- Line 277: Missing a "(" in front of S2 Table.

- Line 279-280: Most common reasons for exclusion "were".

- Line 282: Please standardize spacing before and after "=" as you include spacing here but not elsewhere.

- Lines 331-332: Although I understand the concept of branching logic being maximally harnessed by using electronic data collection, in many low-resource settings this is not a possibility. Is it also appropriate to do this on traditional paper methods?

- Notes section: I might consider moving this to where you discuss Table 2 as otherwise the reader is jumping around quite a bit.

- Standard operating protocols section: This seems a little out of place. I would either cut or elaborate beyond one simple sentence. Perhaps add an example to make this section more relevant and robust.

- Feedback survey: How are the suggestions on the survey incorporated into the common data element set? Are they incorporated immediately or at certain time points? Are the just added in or does consensus have to be achieved on discussion? Perhaps provide some details on this here.

- I may have missed this, but where is the list of the 116 predictors? Perhaps be more clear about where to find these.

Discussion:

- Summary paragraph: This is well written, but it is overall restating a lot of what you have stated above already. I would try to reduce redundancy and streamline a bit if possible here.

- Adoption and use: This paragraph is in future tense, but shouldn't it be past tense since you have already done this? Also, a lot of these 2 paragraphs should go above in the results rather than the discussion. A lot of it is redundant and elaborating on logistics of accessing the Dataverse, so I would say include above where you talk about Dataverse.

Conclusion:

- Final sentence lines 441-444: Instead of saying "This was with the intent..." consider something like "This will enable collaboration", which looks towards the future and proves the benefit of the study.

Reviewer #3: This is well-written and appropriately describes the Delphi process used. The expert consensus group is internationally representative, but is small in number (6 experts). I have no major concerns.

Table 1 is confusing - it might be more helpful to include a sample with variables included, and to describe the significance of bolding and colors in the footnotes. Tables 2 and 3 are more clear.

6. PLOS authors have the option to publish the peer review history of their article (what does this mean?). If published, this will include your full peer review and any attached files.

Reviewer #1: **Yes: **Tapio Niemi

Reviewer #2: No

Reviewer #3: No

---

## [Author Response · Author response to Decision Letter 0]

25 Apr 2021

See "Response to Reviewers" uploaded attachment.

---

## [Decision Letter · Decision Letter 1]

28 May 2021

Common data elements for predictors of pediatric sepsis: A framework to standardize data collection

PONE-D-20-35664R1

Dear Dr. Mawji,

We’re pleased to inform you that your manuscript has been judged scientifically suitable for publication and will be formally accepted for publication once it meets all outstanding technical requirements.

Kind regards,

Valérie Pittet, PhD

Academic Editor

PLOS ONE

Reviewers' comments:

Reviewer's Responses to Questions

**Comments to the Author**

1. If the authors have adequately addressed your comments raised in a previous round of review and you feel that this manuscript is now acceptable for publication, you may indicate that here to bypass the “Comments to the Author” section, enter your conflict of interest statement in the “Confidential to Editor” section, and submit your "Accept" recommendation.

Reviewer #1: All comments have been addressed

2. Is the manuscript technically sound, and do the data support the conclusions?

Reviewer #1: Yes

3. Has the statistical analysis been performed appropriately and rigorously? 

Reviewer #1: N/A

4. Have the authors made all data underlying the findings in their manuscript fully available?

Reviewer #1: Yes

5. Is the manuscript presented in an intelligible fashion and written in standard English?

Reviewer #1: Yes

6. Review Comments to the Author

Reviewer #1: All my comments and suggestions have been taken into account. I have no further concerns and I am happy to recommend the manuscript to be published in Plos One.

7. PLOS authors have the option to publish the peer review history of their article (what does this mean?). If published, this will include your full peer review and any attached files.

Reviewer #1: No

---

## [Editor Report · Acceptance letter]

1 Jun 2021

PONE-D-20-35664R1 

Common data elements for predictors of pediatric sepsis: A framework to standardize data collection 

Dear Dr. Mawji:

I'm pleased to inform you that your manuscript has been deemed suitable for publication in PLOS ONE. Congratulations! Your manuscript is now with our production department. 

Kind regards, 

on behalf of

PD Dr. Valérie Pittet 

Academic Editor

PLOS ONE